# Transferable Adversarial Facial Images for Privacy Protection

### Minghui Li
minghuili@hust.edu.cn
School of Software Engineering,
Huazhong University of Science and
Technology

### Jiangxiong Wang
jiangxiong@hust.edu.cn
School of Software Engineering,
Huazhong University of Science and
Technology

### Hao Zhang
zhanghao99@hust.edu.cn
School of Software Engineering,
Huazhong University of Science and
Technology

### Ziqi Zhou
zhouziqi@hust.edu.cn
School of Computer Science and
Technology, Huazhong University of
Science and Technology

### Shengshan Hu
hushengshan@hust.edu.cn
School of Cyber Science and
Engineering, Huazhong University of
Science and Technology

### Xiaobing Pei
xiaobingp@hust.edu.cn
School of Software Engineering,
Huazhong University of Science and
Technology

## Abstract

The success of deep face recognition (FR) systems has raised serious privacy concerns due to their ability to enable unauthorized tracking of users in the digital world. Previous studies proposed introducing imperceptible adversarial noises into face images to deceive those face recognition models, thus achieving the goal of enhancing facial privacy protection. Nevertheless, they heavily rely on user-chosen references to guide the generation of adversarial noises, and cannot simultaneously construct *natural* and *highly transferable* adversarial face images in black-box scenarios. In light of this, we present a novel face privacy protection scheme with improved transferability while maintain high visual quality. We propose shaping the entire face space directly instead of exploiting one kind of facial characteristic like makeup information to integrate adversarial noises. To achieve this goal, we first exploit global adversarial latent search to traverse the latent space of the generative model, thereby creating natural adversarial face images with high transferability. We then introduce a key landmark regularization module to preserve the visual identity information. Finally, we investigate the impacts of various kinds of latent spaces and find that $\mathcal{F}$ latent space benefits the trade-off between visual naturalness and adversarial transferability. Extensive experiments over two datasets demonstrate that our approach significantly enhances attack transferability while maintaining high visual quality, outperforming state-of-the-art methods by an average 25% improvement in deep FR models and 10% improvement on commercial FR APIs, including Face++, Aliyun, and Tencent.

## CCS Concepts

• **Security and privacy** → **Software and application security**;
**Privacy protections**.

## Keywords

Facial Privacy, Adversarial Example

**ACM Reference Format:**
Minghui Li, Jiangxiong Wang, Hao Zhang, Ziqi Zhou, Shengshan Hu, and Xiaobing Pei. 2024. Transferable Adversarial Facial Images for Privacy Protection. In *Proceedings of the 32nd ACM International Conference on Multimedia (MM '24), October 28-November 1, 2024, Melbourne, VIC, Australia.* ACM, New York, NY, USA, 10 pages. https://doi.org/10.1145/3664647.3681344

## 1 Introduction

Deep face recognition (FR) systems [30, 40] have triumphed in both verification and identification scenarios and been widely applied across various domains, such as security [44], biometrics [28], and criminal-investigation [31]. Despite its promising prospect, FR systems pose a profound risk to individual privacy due to their capacity for large-scale surveillance [2, 46], *e.g.*, tracking user relationship and activities by analyzing face images from social media platforms [13, 36]. Given the opacity of these FR systems, there is an urgent need for an effective black-box approach to protect facial privacy.

Recent works have attempted to protect facial privacy using noise-based adversarial examples (AEs) [32, 38, 49] by adding carefully crafted adversarial perturbations to the source face images to deceive malicious FR systems. However, the adversarial perturbations are typically constrained to the $l_p$ norm within the pixel space, which makes adversarial face images have conspicuous discernible artifacts with poor visual quality [49].

Another solution is exploiting unrestricted adversarial examples [15, 35, 50, 61] to mislead malicious FR systems. Unlike noise-based methods, they are not confined by perturbation budgets, thereby maintaining superior image quality [4, 37, 48]. Recently proposed makeup-based approaches (*e.g.*, AMT-GAN [15], CLIP-2Protect [35]) have achieved state-of-the-art performance, as they can effectively embed adversarial noises in the makeup style by using generative adversarial networks (GANs) [10, 16] or GAN inversion. However, they rely on extra user-chosen guidance, such as reference images in AMT-GAN and textual prompts in CLIP2Protect, to make adversarial noises harmoniously integrated with face characteristics. Besides, these methods excessively focus on the modification of local attributes, leading to minor effects on facial identity characteristics, resulting in limited transferability.

 Minghui Li, Jiangxiong Wang, Hao Zhang, Ziqi Zhou, Shengshan Hu, and Xiaobing Pei

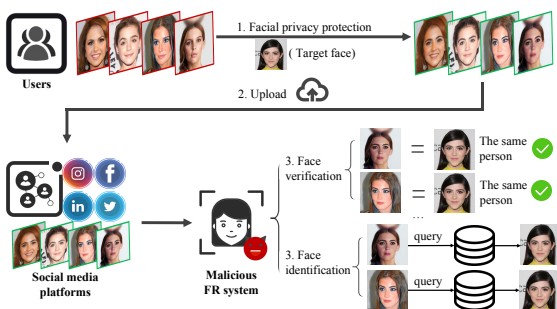

**Figure 1: Illustration of facial privacy protection**

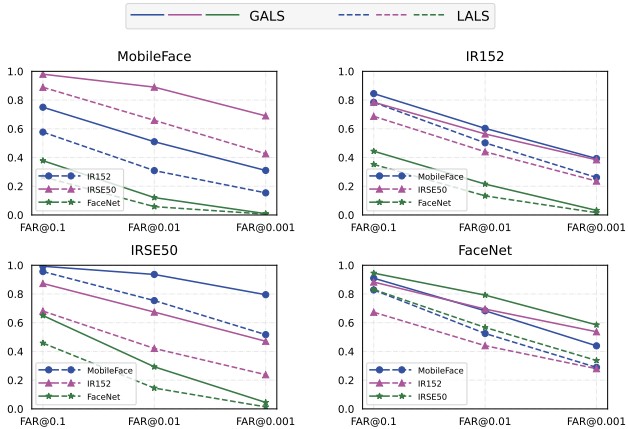

**Figure 2: Evaluation of GALS and LALS on different FR models. We conduct training on a single model and subsequently test it on the remaining three. Results are presented for three different false acceptance rates (*i.e.,* 0.1, 0.01, 0.001).**

To address the above issues, we propose the **G**uidance-**I**ndependent Adversarial **F**acial Images with **T**ransferability (*GIFT*) for privacy protection, which takes a big step towards bridging the gap between visual naturalness and adversarial transferability. Firstly, we map face images to a low-dimensional manifold represented by a generative model. We then conduct adversarial latent optimization that moves along the adversarial direction. In contrast to [35], we perform adversarial latent optimization over the global latent space called *Global Adversarial Latent Search (GALS)*, which can control more semantic information with improved transferability. However, in the absence of extra guidance information, GALS may change the visual identity of resulting images. Therefore, we introduce a *key landmark regularization (KLR)* method to rectify this issue. Furthermore, we investigate the effect of diverse latent spaces on our scheme. We find that the latent space $\mathcal{W}^+$, which is commonly used in existing face privacy protection tasks, exhibits weaker transferability and lower perceptual image quality than the other two prevalent latent spaces $\mathcal{W}$ and $\mathcal{F}$. Consequently, we opt for the *optimal latent space* $\mathcal{F}$ to further enhance our design. In summary, our main contributions include:

- We propose a novel facial privacy protection approach using Global Adversarial Latent Search to construct natural and highly transferable adversarial face images without extra guidance information.
- We reveal the limitations of $\mathcal{W}^+$ latent space and the intriguing properties of the other two prevalent latent spaces $\mathcal{W}$ and $\mathcal{F}$ under the facial privacy protection scenario.
- Extensive experiments on both face verification and identification tasks demonstrate the superiority of our approach against various deep FR models and commercial APIs. Notably, we achieved a significant improvement of 25% than existing schemes in terms of transferability.

## 2 Related Work and Background

### 2.1 Facial Privacy Protection

Recently, many works have been proposed to protect facial privacy against unauthorized FR systems [29, 42, 43]. The typical strategy involves the utilization of noise-based adversarial examples [29, 49, 57–60], where carefully crafted perturbations are added to face images to deceive malicious FR models. Oh *et al.* [29] proposed crafting protected face images from a game theory perspective in the white-box setting, which is impractical in real-world scenarios.

Thus TIP-IM [49] introduced the idea of generating adversarial identity masks in the black-box setting. However, the perturbations are usually perceptible to humans and affect the user experience.

Another strategy is to leverage unrestricted adversarial examples [4, 18, 37, 48, 52, 56], which are not constrained by the perturbation norm in the pixel space and enjoy a better image quality [4, 37, 48]. Among these, makeup-based unrestricted adversarial examples are presented against unknown FR systems by concealing adversarial perturbations within natural makeup characteristics. Zhu *et al.* [61] made the first effort to utilize makeup to generate protected face images in the white-box setting. Afterwards, Adv-Makeup [50] synthesized imperceptible eye shadow over the orbital region on the face, which has limited transferability. AMT-GAN [15] generated adversarial face images with makeup transferred from reference images in a black-box manner, which has a higher attack success rate but suffers from obvious artifacts due to the conflict between the makeup transfer module and the adversarial noises. Recently, CLIP2Protect [35] traversed over the local latent space that controls the makeup style of a pre-trained generative model by using text prompts. However, all the above methods rely on guidance (*e.g.*, text or image references) to make the adversarial noises distributed in a natural way. This is a disappointing constraint in real-world applications as the users usually have no desired target references. More importantly, the visual quality of output face images is largely affected by the references, as detailedly discussed in Section 3.2. DiffProtect [26] and Adv-Diffusion [24] employ the diffusion models [53] as the generative models and iteratively refine the latent representations of facial images, although yielding higher-quality adversarial facial images, exhibit limited attack transferability, rendering it impractical for real-world applications.

### 2.2 GAN Inversion

GAN inversion [1, 19, 39, 41, 47] intends to invert a given face image back to a low-dimensional manifold which is expressed as a latent space of a pre-trained GAN model, such that the image can be

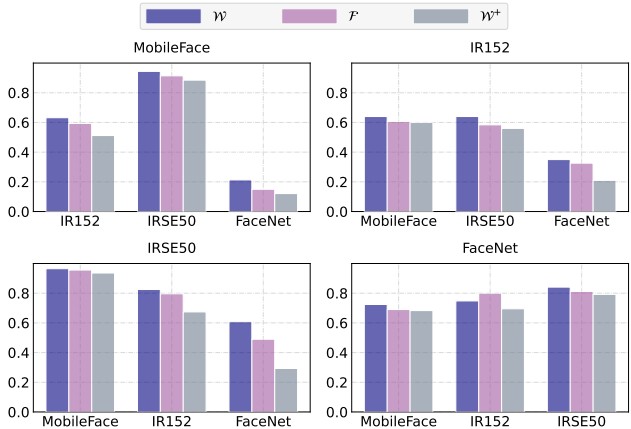

Figure 3: Protection success rates of different latent spaces on four FR models in the black-box setting. Specifically, we perform training on a single model and subsequently test it on the remaining three. We set the false match rate of 0.01 for each model.

faithfully reconstructed. As the StyleGAN [22] models trained on a high-resolution face image dataset [21] exhibit exceptional image synthesis capabilities, various GAN inversion methods have been developed using different latent spaces based on StyleGANs. Generally, there are three typical latent spaces (*i.e.*, $\mathcal{W}$ [1], $\mathcal{W}^+$ [39], and $\mathcal{F}$ [19]). They are the trade-off design between the reconstruction quality and editability [25]. $\mathcal{W}$ uses a mapping network to disentangle different features with a high degree of editability. However, it has limited expressiveness which restricts the range of images that can be faithfully reconstructed. Meanwhile, $\mathcal{W}^+$ feeds different intermediate latent vectors into each layer of the generator via AdaIN [17], alleviating the image distortion at the expense of editability. The latent space $\mathcal{F}$ consists of specific features which enjoy the highest reconstruction quality but suffer from the worst editability.

## 3 Methodology

### 3.1 Problem definition

In general, FR systems can operate with two modes: *face verification* and *face identification*. For verification, the FR systems identify whether two face images correspond to the same identity. For identification, the FR systems query the face database to identify whose representation is closest to the input image. In this paper, we consider both scenarios to sufficiently demonstrate the effectiveness of our approach. As seen in Fig. 1, if the user's source face image $x_s$ is directly posted to social media platforms, malicious FR systems could potentially trace the relationships and activities of the user by analysing the publicly available images.

With the help of facial privacy protection algorithms, users can obtain the protected face image $x_p$ that appears indistinguishable to human (*i.e.*, naturalness) but can deceive the FR systems (*i.e.*, transferablility). For the malicious FR systems, $x_p$ has the same identity as the target impersonated image $x_t$ in both face verification and face identification. Generally, the problem can be formulated as:

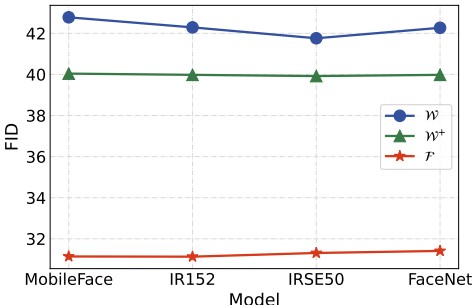

Figure 4: FID comparison in three latent spaces

$$\min_{x_p} \mathcal{L}_{adv} = \mathcal{D}\left(f_n\left(x_p\right), f_n\left(x_t\right)\right)$$
$$\text{s.t. } \mathcal{H}\left(x_p, x_s\right) \le \epsilon \tag{1}$$

where $\mathcal{D}\left(\cdot\right)$ represents a distance metric and $f_n\left(\cdot\right)$ stands for a FR model that outputs a feature vector by extracting the feature representation of a face image. Contrary to noise-based adversarial examples where $\mathcal{H}\left(x_p, x_s\right) = \left\|x_s - x_p\right\|_p$ and $\|\cdot\|_p$ is the $L_p$ norm, $\mathcal{H}\left(x_p, x_s\right) \le \epsilon$ quantifies the extent of unnaturalness of $x_p$ compared to $x_s$.

### 3.2 Challenges and limitations

Existing works [15, 35] have experimentally demonstrated that the guidance of the reference image or text prompts plays an important role in maintaining the visual quality of adversarial face images during the process of generation. Without using guidance, the adversarial perturbations will be randomly generated and spread all over the face without any constraints, leading to obvious artifacts in the result images. If we simply fix the adversarial area, the noises may make the modified area extremely strange and distorted. Therefore, existing works have to use extra information to guide the distribution of adversarial noises such that they can harmonize with facial characteristics.

Moreover, even with makeup information guidance, due to the incomplete disentanglement of facial attributes, the adversarial modifications concentrating on one kind of local semantics struggle to integrate seamlessly with other facial semantic information. As shown in Fig. 7, the visual quality can still be damaged in guidance-based schemes, especially when the reference is not appropriately chosen. Note that although CLIP2Protect [35] maintains the face quality well in some cases, the background behind the face has been significantly damaged. It should be emphasized that the image's visual quality should contain not only the facial information but also the background.

Besides, these methods excessively focus on local attributes, leading to minor effects on facial identity characteristics. This limitation results in limited transferability and renders them less practical for real-world applications.

### 3.3 Our key insights

To address the aforementioned issues, we propose directly manipulating the entire facial space to harmoniously integrate adversarial noises, rather than solely using one kind of facial characteristic to

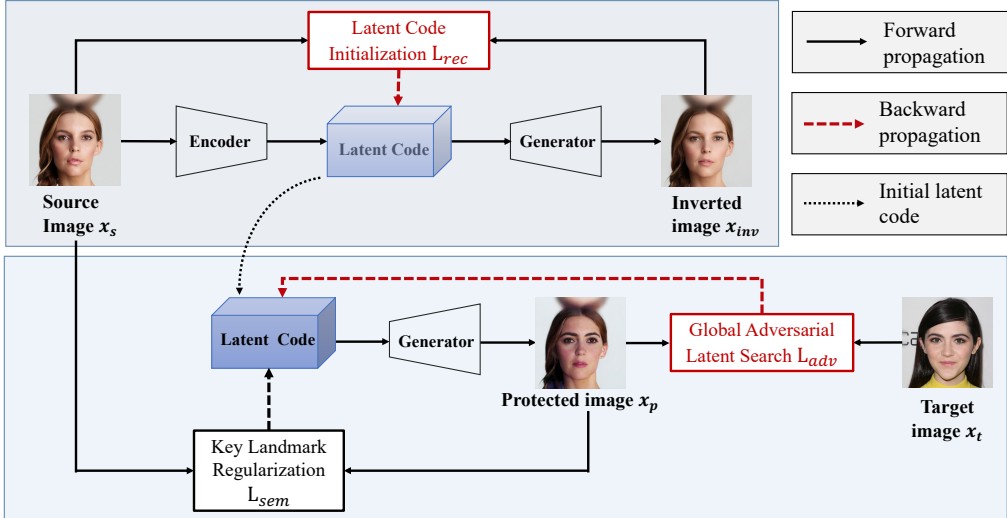

**Figure 5: The framework of GIFT**

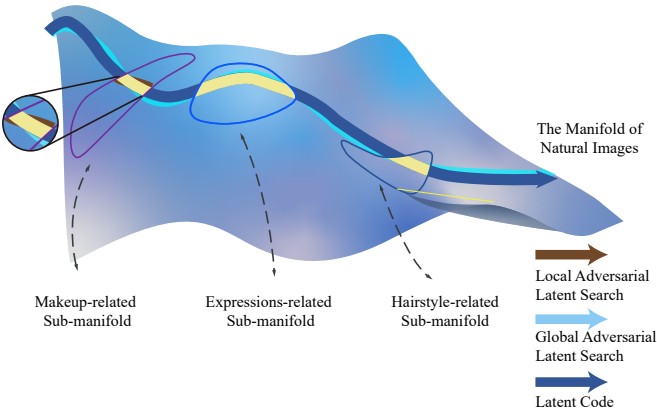

**Figure 6: Differences between Global Adversarial Latent Search and Local Adversarial Latent Search. Global Adversarial Latent Search seamlessly modifies the latent code along the adversarial direction. In contrast, LALS merely adjust the local attributes, which has many limitations.**

guide the distribution of noises. In this way, all of the facial information will be adaptively assembled, including makeup characteristics, expressions, and face shape, as well as adversarial noises. In addition, by comprehensively optimizing every feature rather than excessively optimizing a particular attribute, the transferability of generated adversarial facial images is significantly improved. To this end, we first have the following observations.

**Observation I: Global adversarial latent search exhibits enhanced transferability.** In order to maintain the naturalness of the face images $x_p$, CLIP2Protect [35] only optimizes the latent codes corresponding to the *deep layers* of StyleGAN that is associated with the makeup style. This method is essentially a kind of *Local Adversarial Latent Search (LALS)*. Meanwhile, numerous studies [3, 23] have affirmed that the *initial layers* of StyleGAN

control more face image attributes, such as pose, hairstyle, and face shape. Therefore, we speculate that optimizing both the deep layers and initial layers of latent codes, *a.k.a, Global Adversarial Latent Search (GALS)*, is more beneficial to enhance the adversarial transferability, as they can control more semantic information.

Fig. 6 provides a visualization example to illustrate this observation. Specifically, LALS only focuses on optimizing local single attributes, such as makeup-related features, potentially disturbing adjacent attribute distributions or those that are not incompletely decoupled, which might impact the visual effect. Moreover, the modification of one single attribute is limited, resulting in relatively smaller repercussions on the overall facial identity features compared to GALS. Conversely, GALS harmoniously refines the entire facial characteristics to sophisticatedly guide the distribution of noises along the adversarial direction in the low-dimensional manifold [6] represented as the latent space. We perform experiments to validate this observation, as shown in Fig. 2, and find that GALS has a higher protection success rate compared to LALS in the black-box setting, indicating far better transferability than GALS.

**Observation II: Key landmark regularization helps ensure visual identity.** Due to the absence of guidance information, our approach may result in changes of visual identity during the process of global optimization. Therefore, there is a need for a new regularization method to maintain visual identity. Intuitively, leveraging the traditional MSE is promising for aligning the protected image $x_p$ with the source image $x_s$. Nevertheless, pixel-level regularization may lead to the emergence of artifacts in the face image and a reduction of adversarial effectiveness. Inspired by [26], we introduce a method called *key landmark regularization (KLR)* due to their capability to achieve region-level regularization, which aligns two semantic segmentation maps of the protected image $x_p$ and the source image $x_s$ in the optimization process. This method ensures consistency in the distribution of facial characteristics and the overall face shape between $x_s$ and $x_p$.

---

**Algorithm 1:** Transferable Adversarial Facial Images

---

**Input:** Source image $x_s$, target image $x_t$, generator $G$,
      encoder $E_b$, semantic encoder $E_s$, optimizer $Adam$.
**Parameter:** Iterations $T_1$, iterations $T_2$, transformation
      probability $p$, hyper-parameters $\lambda_{per}, \lambda_{sem}$.
**Output:** Protected image $x_p$.

1  Generate initial latent code $w_{ini} = E_b(x_s)$ ;
2  **for** $i = 0$ *to* $T_1 - 1$ **do**
3      Calculate $\mathcal{L}_{rec}$ with Eq. (2);
4      $w_{ini} \leftarrow Adam(w_{ini}, \mathcal{L}_{rec})$;
5  **end**
6  Obtain the initialized latent code $w_f = w_{ini}$;
7  **for** $i = 0$ *to* $T_2 - 1$ **do**
8      Obtain intermediate image $x_{pi} = G(w^f)$;
9      Calculate semantic maps $sem_s = E_s(x_s)$ and
        $sem_p = E_s(x_{pi})$;
10     Calculate $\mathcal{L}_{total}$ with Eq. (5);
11     $w^f \leftarrow Adam(w^f, \mathcal{L}_{total})$;
12  **end**
13  Obtain final latent code $w_o^f$;
14  **Return the protected image** $x_p = G(w_o^f)$.

---

**Observation III: $\mathcal{F}$ latent space benefits the trade-off between visual naturalness and adversarial transferability.** It is critical for a GAN inversion method to choose a good latent space that can reconstruct the face images faithfully and facilitate downstream tasks. Unfortunately, the latent space $\mathcal{W}^+$ widely used in the existing facial privacy protection schemes has poor reconstruction quality [25]. In light of this, we explore the performance of the other two typical latent spaces, $\mathcal{W}$ and $\mathcal{F}$. As shown in Fig. 3 and Fig. 4, the latent space $\mathcal{F}$ not only exhibits the best image quality but also demonstrates strong black-box attack success rate in the black-box setting, achieving the optimal trade-off between naturalness and high adversarial transferability.

### 3.4 Transferable Adversarial Facial Images

Drawing inspiration from the above analysis, we propose a brand-new generative framework (GIFT) to generate guidance-independent adversarial facial images with transferability for facial privacy protection. The pipeline of our approach is depicted in Fig. 5. We first employ GAN inversion to initialize the latent code that can faithfully reconstruct the source face image in $\mathcal{F}$ rather than $\mathcal{W}^+$ [35] latent space. Then, we conduct the global adversarial latent search, which takes the protected image and the target image as inputs to adversarially optimize the latent code of the protected face image. Furthermore, we introduce a key landmark regularization to preserve the visual identity of the protected image.

**Latent Code Initialization:** The latent code is initialized based on GAN inversion. Given a source image $x_s$ and pre-trained encoder $E_b$ from [19], we first calculate the initial latent code $w_{ini} = E_b(x_s)$ in the $\mathcal{F}$ latent space. Then we optimize the latent code $w^{ini}$ by a reconstruction loss, which is defined as:

$$
\begin{aligned}
\mathcal{L}_{rec}\left(w^{ini}\right) &= \mathcal{L}_{mse}\left(w^{ini}\right) + \alpha \mathcal{L}_{per}\left(w^{ini}\right) \\
&= \left\| x_s - G\left(w^{ini}\right) \right\|^2 \\
&+ \alpha \left\| F(x_s) - F\left(G\left(w^{ini}\right)\right) \right\|^2
\end{aligned}
\tag{2}
$$

where $G$ refers to the pre-trained generative model, $\mathcal{L}_{mse}$ and $\mathcal{L}_{per}$ denote mean-squared-error (MSE) and perceptual loss, respectively. $\alpha$ is the weight assigned to $\mathcal{L}_{per}$. $F(\cdot)$ represents an LPIPS [55] network used to compute the perceptual distance. As a result, we obtain the initialized latent code $w_f$, which can faithfully reconstruct $x_s$ by $G$.

**Global Adversarial Latent Search:** We utilize an ensemble training strategy with input diversity to search for a good adversarial optimization direction similar to [15]. We select $N$ pre-trained FR models $\{f_n(\cdot)\}_{n=1}^N$ which exhibit high accuracy in the public facial datasets, serving as white-box models to imitate the decision boundaries of potential target models in the black-box setting during performing global optimization. The adversarial loss is:

$$
\mathcal{L}_{adv} = \frac{1}{N} \sum_{n=1}^N \mathcal{D}\left( f_n\left(T\left(G\left(w^f\right), p\right)\right), f_n(x_t) \right)
\tag{3}
$$

where $\mathcal{D}(x_1, x_2) = 1 - \cos(x_1, x_2)$ is the cosine distance, $f_n(\cdot)$ represents the $n$-th local pre-trained white-box model which maps an input face image to a feature representation. $T(\cdot)$ represents the transformation function including image resizing and Gaussian noising, and $p$ is a predefined probability that determines whether the transformation will be applied to $G(w^f)$.

**Key Landmark Regularization:** We first leverage a pre-trained semantic encoder from [51] to obtain the face semantic segmentation maps $sem_s$ and $sem_p$ for the source and protected image. We then take advantage of the semantic segmentation maps to regularize the optimization process, ensuring that the protected image preserves visual identity for humans. The regularization loss is defined as:

$$
\mathcal{L}_{sem} = \mathcal{L}_{CE}\left( \mathcal{M}\left(G\left(w^f\right)\right), \mathcal{M}(x_s) \right)
\tag{4}
$$

where $\mathcal{L}_{CE}(\cdot)$ is the cross-entropy loss, $\mathcal{M}(\cdot)$ represents the pre-trained semantic encoder. The total loss is:

$$
\mathcal{L}_{total} = \lambda_{adv} \mathcal{L}_{adv} + \lambda_{sem} \mathcal{L}_{sem}
\tag{5}
$$

where $\lambda_{adv}$ and $\lambda_{sem}$ represent the hyper-parameters. The whole optimization process is outlined in Algorithm. 1.

## 4 Experiments

### 4.1 Experiments Setup

**Implementation Details:** We employ StyleGAN2 pre-trained on FFHQ [21] as our generative model and use the Adam optimizer in all experiments. For Latent Code Initialization, we iteratively optimize the latent code for 1200 steps with a learning rate of 0.01. During the adversarial optimization process, we traverse over the global latent space for 50 iterations with a learning rate of 0.002 to generate protected face images. We set the hyper-parameters $\alpha$, $\lambda_{adv}$ and $\lambda_{sem}$ to 10, 1, 0.01, respectively.

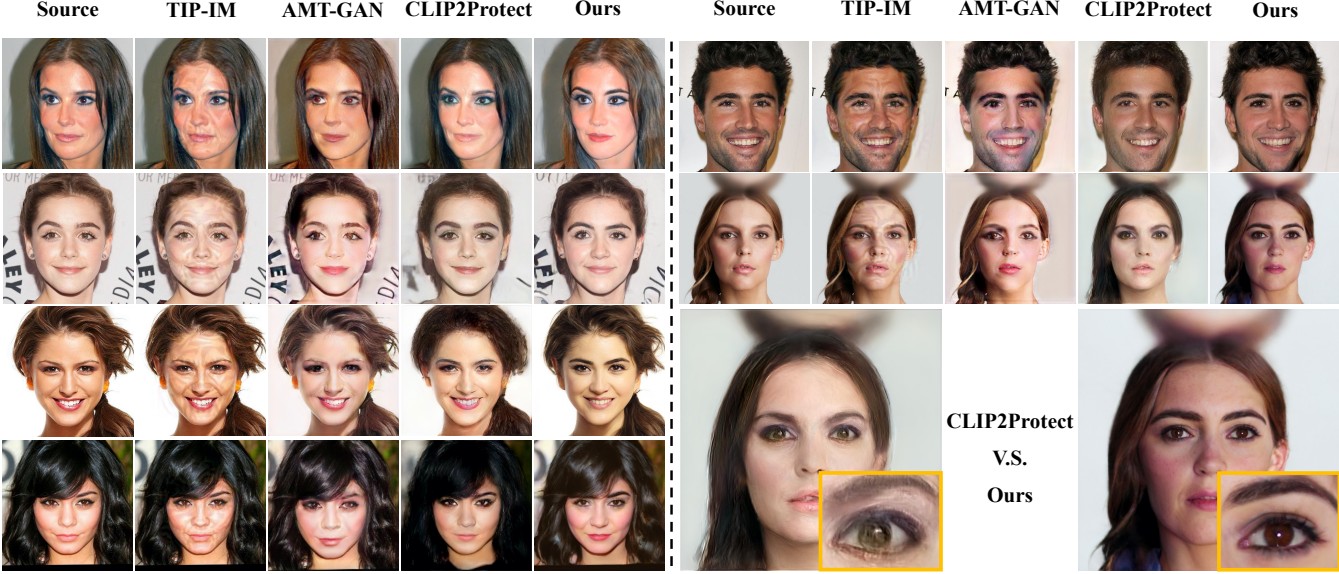

**Figure 7: Comparion of visual quality between SOTA noise-based TIP-IM, unrestricted AMT-GAN, CLIP2Protect and our method**

**Table 1: Protection success rate (%) of impersonation attack under the face verification task**

| Method | CelebA-HQ | | | | LADN-Dataset | | | | Average |
|---|---|---|---|---|---|---|---|---|---|
| | IRSE50 | IR152 | FaceNet | MobileFace | IRSE50 | IR152 | FaceNet | MobileFace | |
| Clean | 7.29 | 3.80 | 1.08 | 12.68 | 2.71 | 3.61 | 0.60 | 5.11 | 4.61 |
| PGD [27] | 36.87 | 20.68 | 1.85 | 43.99 | 40.09 | 19.59 | 3.82 | 41.09 | 25.60 |
| MI-FGSM [8] | 45.79 | 25.03 | 2.58 | 45.85 | 48.90 | 25.57 | 6.31 | 45.01 | 30.63 |
| TI-DIM [9] | 63.63 | 36.17 | 15.30 | 57.12 | 56.36 | 34.18 | 22.11 | 48.30 | 41.64 |
| Adv-Makeup(IJCAI'21) [50] | 21.95 | 9.48 | 1.37 | 22.00 | 29.64 | 10.03 | 0.97 | 22.38 | 14.72 |
| TIP-IM(ICCV'21) [49] | 54.40 | 37.23 | 40.74 | 48.72 | 65.89 | 43.57 | 63.50 | 46.48 | 50.06 |
| AMT-GAN(CVPR'22) [15] | 76.96 | 35.13 | 16.62 | 50.71 | 89.64 | 49.12 | 32.13 | 72.43 | 52.84 |
| CLIP2Protect(CVPR'23) [35] | 81.10 | 48.42 | 41.72 | 75.26 | 91.57 | 53.31 | 47.91 | 79.94 | 64.90 |
| GIFT (Ours) | **95.70** | **92.50** | **63.50** | **91.90** | **94.61** | **98.20** | **85.03** | **96.41** | **89.73** |

**Table 2: Protection success rate (%) of impersonation attacks under the face identification task**

| Method | IRSE50 | | IR152 | | FaceNet | | MobileFace | | Average | |
|---|---|---|---|---|---|---|---|---|---|---|
| | R1-T | R5-T | R1-T | R5-T | R1-T | R5-T | R1-T | R5-T | R1-T | R5-T |
| TIP-IM [49] | 16.2 | 51.4 | 21.2 | 56.0 | 8.1 | 35.8 | 9.6 | 24.0 | 13.8 | 41.8 |
| CLIP2Protect [35] | 24.5 | 64.7 | 24.2 | 65.2 | 12.5 | 38.7 | 11.8 | 28.2 | 18.2 | 49.2 |
| GIFT (Ours) | **69.8** | **93.6** | **72.0** | **87.6** | **44.8** | **70.2** | **41.6** | **80.2** | **57.1** | **82.9** |

**Table 3: Comparison of FID and PSR Gain. PSR Gain is absolute gain in PSR relative to Adv-Makeup.**

| Method | FID ↓ | PSR Gain ↑ |
|---|---|---|
| Adv-Makeup [50] | 4.23 | 0 |
| TIP-IM [49] | 38.73 | 35.34 |
| AMT-GAN [15] | 34.44 | 38.12 |
| CLIP2Protect [35] | 46.34 | 50.18 |
| GIFT (Ours) | **31.19** | **75.01** |

**Datasets:** We perform experiments for both face verification and identification tasks. *Face Verification*: We utilize CelebA-HQ [20] and LADN [11] as our test set. Specifically, for CelebA-HQ, we select a subset which contains 1000 images with different identities. For LADN, we divide the 332 images into 4 groups, with each group of images impersonating a target identity provided by [15]. *Face Identification*: we randomly select 500 images of 500 different identities in CelebA-HQ as the probe set, and the corresponding 500 images of the same identities along with the target image $x_p$ to form the gallery set.

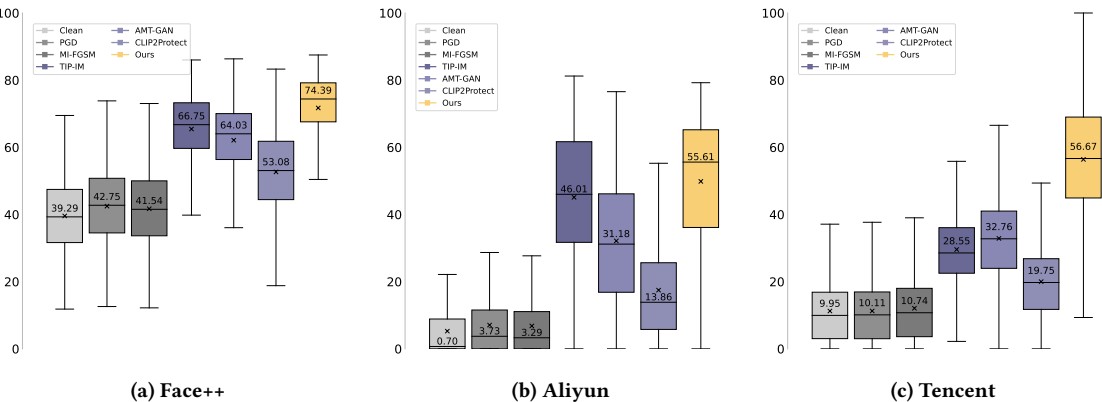

**Figure 8: Confidence scores returned from commercial APIs**

**Target Models:** Following [15], we choose various deep FR models and commercial FR APIs to evaluate the transferability of the adversarial facial images generated by GIFT in the black-box settings. Specifically, the deep FR models include MobileFace [5], IR152 [7], IRSE50 [14], and FaceNet [33] and commercial FR APIs include Face++, Aliyun, and Tencent.

**Competitors:** We compare GIFT with recent noise-based and makeup-based facial privacy protection approaches. Noised-based methods include PGD [27], MI-FGSM [8], TI-DIM [9], and TIP-IM [49]. Makeup-based approaches include Adv-Makeup [50], AMT-GAN [15] and CLIP2Protect [35]. Among these methods, TIP-IM and CLIP2Protect are regarded as the state-of-the-art (SOTA) approaches against *black-box* FR systems in noise-based and unrestricted settings, respectively. Notably, TIP-IM employs a multi-target objective within its optimization to discover the best image from multiple targets. To maintain fairness in the comparison, we employ a single-target variant.

**Evaluation Metrics:** We employ different evaluation strategies to calculate the protection success rate (PSR) for the verification and identification scenarios. For verification, we identify that two face images belong to the same identity if $\mathcal{D}\left(x_p, x_s\right) \geq \tau$ and then calculate the proportion of successfully protected images in relation to all images. For identification, we report the Rank-N targeted identity success rate, which means that at least one of the top N images belongs to the target identity after ranking the distance for all images in the gallery to the given probe image. For commercial FR APIs, we directly record the confidence scores returned by FR servers. We also leverage FID [12], PSNR (dB) and SSIM [45] to evaluate the image quality. The FID quantifies the dissimilarity between two data distributions [54], commonly employed to assess the extent to which a generated dataset resembles one obtained from the real world. PSNR and SSIM are commonly utilized techniques for assessing the difference between two images.

## 4.2 Comparison Study

**Evaluation on Black-box FR Models.** We present experimental results of GIFT in the black-box settings on four different pretrained FR models on two public datasets under face verification and identification tasks. For face verification, we set the system threshold value at 0.01 false match rate for each FR model *i.e.*, IRSE50

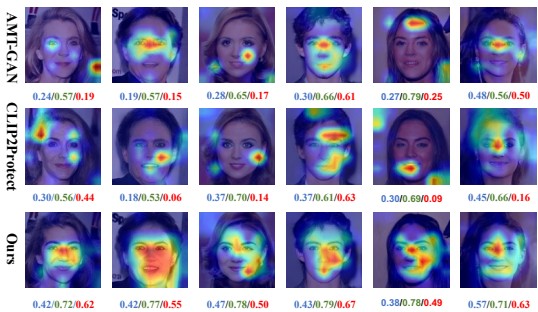

**Figure 9: Visualization of gradient response using Grad-CAM on FR model (IRSE50). The numbers under each image represent their cosine similarity with the target image, along with the confidence scores from the commercial APIs (Face++, Aliyun).**

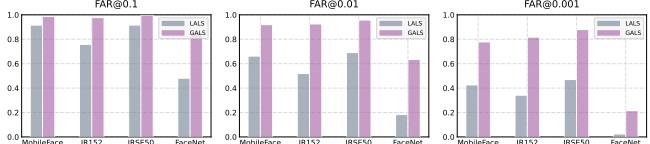

**Figure 10: PSR comparison of GALS and LALS**

(0.241), IR152 (0.167), FaceNet (0.409), and MobileFace (0.302). The quantitative results in terms of PSR under the face verification scenario are displayed in Tab. 1. The results show that GIFT has the capability to achieve an average absolute gain of about 25% and 39% over SOTA unrestricted and noise-based facial privacy protection methods, respectively. We also provide PSR under the face identification scenario in Tab. 2. Consistently, GIFT outperforms recent methods in both Rank-1 and Rank-5 settings. Given that AMT-GAN and Adv-Makeup were initially trained for impersonating the target identity in the verification task, they have not been incorporated into Tab. 2.

**Evaluation on Image Quality.** Tab. 3 shows the quantitative evaluations on image quality. Notably, we report the results that

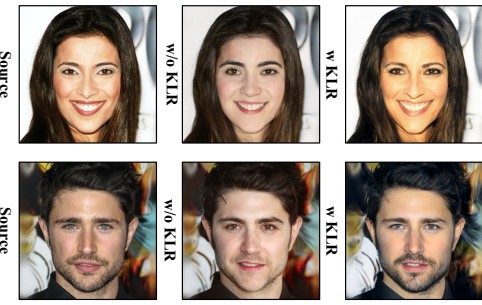

**Figure 11: The effect of key landmark regularization**

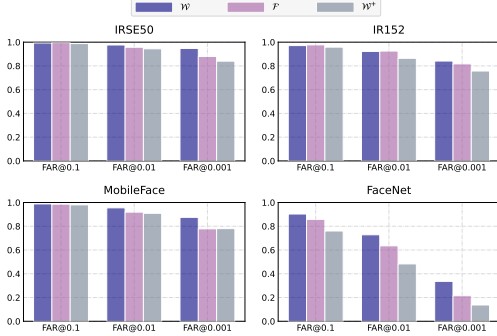

**Figure 12: PSR comparison of different latent spaces**

**Table 4: FID comparison of different latent spaces**

| Latent space | MobileFace | IR152 | IRSE50 | FaceNet | Average |
|:---:|:---:|:---:|:---:|:---:|:---:|
| $\mathcal{W}$ | 42.80 | 41.78 | 42.51 | 41.49 | 42.14 |
| $\mathcal{W}^+$ | 40.05 | 39.92 | 39.99 | 39.91 | 39.97 |
| $\mathcal{F}$ | 31.15 | 31.12 | 31.13 | 31.42 | 31.26 |

are averaged over 10 text prompts for CLIP2Protect. Although Adv-Makeup [50] achieves the lowest FID score, its transferability is limited due to perturbing only the eye region. Beyond Adv-Makeup, GIFT yields better FID results with the highest transferability, indicating that the adversarial face images generated by GIFT are more natural.

In addition, we give a qualitative comparison of visual image quality in Fig. 7 with noise-based method TIP-IM and makeup-based unrestricted methods AMT-GAN and CLIP2Protect. TIP-IM's adversarial face images contain noticeable noise that can be easily perceived by humans. The makeup generated by AMT-GAN sometimes does not align well with the facial characteristics, resulting in artifacts on the face images. The adversarial face images generated by CLIP2Protect have an obvious "painted-on" appearance, especially when viewed at high resolution. In contrast, GIFT produces adversarial face images that look natural and effectively preserve the background content, making it applicable in practical scenarios.

**Evaluation on Commercial APIs.** We further demonstrate the effectiveness of GIFT on commercial APIs (*i.e.*, Face++, Aliyun, and Tencent) in face verification mode. The output of these APIs is a confidence score on a scale of 0 to 100, serving as a metric

for the similarity between two images. A higher score signifies a greater degree of similarity. The training data and model parameters of these commercial APIs are undisclosed, effectively simulating scenarios in real-world face privacy protection. We present the average confidence scores for 1000 images from the CelebA-HQ dataset in Fig. 8, demonstrating the superiority of GIFT over other competitors.

### 4.3 Visualization and Analysis

In this section, we employ Grad-CAM [34] to explore why the adversarial face images generated by GIFT exhibit better transferability. We present gradient visualizations of adversarial face images generated by AMT-GAN, CLIP2Protect, and GIFT in the black-box setting compared to the target images in Fig. 9. We can observe that the gradient responses of AMT-GAN and CLIP2Protect either concentrate on specific facial regions or focus on the background of the face image. In contrast, the gradient responses of GIFT are concentrated on the facial region, without being limited to local features, but rather covering the entire face. Therefore, the adversarial face images generated by GIFT exhibit superior performance, both on deep FR models and commercial FR APIs.

### 4.4 Ablation Study

**The Effect of KLR:** We investigate the effect of key landmark regularization on GIFT. As shown in Fig. 11, with the absence of KLR, the visual identity of the protected face image will change.

**The Effect of GALS:** We analyze the effect of GALS on GIFT. As depicted in Fig. 10, compared to LALS, which only adversarially modifies the makeup style, GALS exhibits significantly better transferability.

**The Effect of Latent Space:** We study the effect of different latent spaces on GIFT. Both Fig. 12 and Tab. 4 indicate that, although $\mathcal{W}$ latent space results in better adversarial transferability, the generated adversarial face images exhibit poor image quality. In contrast, $\mathcal{F}$ latent space maintains high transferability while achieving the best image quality.

### 5 Conclusion

In this paper, focusing on protecting facial privacy against malicious FR systems, we propose GIFT, a guidance-independent generative framework to construct highly transferable adversarial facial images while maintain good visual effect. Specifically, we leverage Global Adversarial Latent Search to construct natural and highly transferable adversarial face images without extra guidance information. We further reveal the limitations of $\mathcal{W}^+$ latent space and the intriguing properties of the other two prevalent latent spaces $\mathcal{W}$ and $\mathcal{F}$ under the facial privacy protection scenario. Extensive experiments on both face verification and identification tasks demonstrate the superiority of GIFT against various deep FR models and commercial FR APIs.

### Acknowledgments

Minghui's work is supported in part by the National Natural Science Foundation of China (Grant No.62202186). Shengshan's work is supported in part by the National Natural Science Foundation of China (Grant No.62372196). Ziqi Zhou is the corresponding author.

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
