# OpenReview forum: "Transferable Adversarial Facial Images for Privacy Protection"
_acmmm.org/ACMMM/2024/Conference — MM2024 Poster_

### Official Review · Reviewer_gFB4 · 2024-05-13

**Rating:** 4
**Confidence:** 3

**Summary:**

The paper proposes to use adversarial perturbation to protect facial privacy by mislead the commercial face recognition systems to mis-identify their ids.

**Strengths:**

The ASR rate is higher than quite a few benchmark approaches.

**Limitations:**

The main performance lift is driven by the global adversarial search. However, it is unclear how many white-box models are used to designed the adversarial loss. In addition, there lacks an ablation study to show how the performance varies when different models are used.
Figure 11 is strange since it is difficult to understand why the shape regularizer can help to generate the beard.
Figure 6 is less useful which can only introduce confusion to the main idea.
It is also better to clarify whether the encoder is fixed or can be fine-tuned.
Overall, the approach achieves good results but contribution is incremental.

**Suitability:**

2

---

### Official Review · Reviewer_9LUt · 2024-05-26

**Rating:** 4
**Confidence:** 3

**Summary:**

This paper presents GIFT, a novel face privacy protection scheme that enhances the transferability of adversarial face images while maintaining high visual quality. Unlike previous methods that require user-chosen references, GIFT utilizes Global Adversarial Latent Search (GALS) to directly traverse the latent space of generative models, generating natural adversarial images. A key landmark regularization module is introduced to preserve visual identity. Extensive experiments show that GIFT outperforms existing methods by 25% in deep face recognition models and 10% on commercial APIs, demonstrating superior attack transferability and visual fidelity.

**Strengths:**

1. The proposed method exhibits enhanced transferability which achieves 25% improvement in deep face recognition and 10% in commercial APIs, ensuring more effective adversarial images.
2. GIFT maintains natural and high-fidelity adversarial images, preserving original facial identity and suitable for real-world applications.
3. Require no extra guidance information, making it more user-friendly and practical for generating adversarial images.
4. Insightful observations are presented.

**Limitations:**

1. The proposed framework is kind of interesting but not novel enough.
2. The framework figure in the paper is simple and lacks a detailed caption.
3. The ablation about $\lambda_{adv}$ and $\lambda_{sem}$ is absent.

**Suitability:**

2

---

### Official Review · Reviewer_nXL7 · 2024-06-08

**Rating:** 3
**Confidence:** 3

**Summary:**

This paper presents a privacy protection method based on adversarial facial images. The authors thoroughly explore the impact of different latent spaces on protection success rates, including global/local latent spaces, $F/W/W^+$ latent spaces, ultimately achieving a significant improvement in protection success rates compared to existing methods. Additionally, the authors propose a key landmark regularization method that effectively maintains image quality.

**Strengths:**

The proposed method significantly surpasses existing work in terms of protection success rate, while also maintaining high image quality.

**Limitations:**

1. The novelty is limited. The authors propose two contributions to improve the protection success rate: using the $F$ latent space instead of the $W^+$ latent space, and additionally optimizing the initial layers of StyleGAN. Clearly, both of these contributions are very incremental, resembling the adjustment of two hyperparameters on existing work.
2. There is no comparison of the runtime efficiency of different algorithms. Section 4.1 shows that 1200 iterations are required for the latent code initialization step, which seems to account for the majority of the total runtime of the proposed method. How many iterations do other methods require?
3. The results of the ablation study are not fully reported. For the role of KLR, the authors show the image quality with and without KLR, but do not report the protection success rate. For GALS, the authors report the protection success rates of GALS and LALS, but do not report the FID.
4. Many hyperparameters lack the corresponding ablation study, including $\alpha, \lambda_{adv}, \lambda_{sem}$.
5. The organization of the paper is unreasonable. Figures 3 and 4 have already shown the effects of different latent spaces, which makes Figure 12 in Section 4.4 have no new informational value.
6. Section 4.1 mentions that more details is in the supplementary material, but I couldn't find it.

**Suitability:**

2

---

### Meta-Review · Area_Chair_BzHh · 2024-07-02

**Recommendation:** Accept (Poster)
**Confidence:** 5

**Metareview:**

This paper introduces a new face privacy protection scheme that improves the transferability of adversarial face images while preserving high visual quality. The idea is intriguing and the writing in the paper is clear. The proposed method, GIFT, demonstrates its effectiveness by achieving a 25% improvement in deep face recognition and a 10% improvement in commercial APIs. After carefully reviewing the paper and taking into account the feedback from reviewers, I recommend accepting this paper. It has the potential to provide valuable insights to the face privacy protection community. I encourage the authors to refine their work based on suggestions from all reviewers to create an even stronger version of their research.